# Unsupervised Keypoint Learning
# for Guiding Class-Conditional Video Prediction

**Yunji Kim[1], Seonghyeon Nam[1], In Cho[1], and Seon Joo Kim[1,2]**
[1]Yonsei University        [2]Facebook
{kim_yunji,shnnam,join,seonjookim}@yonsei.ac.kr

## Abstract

We propose a deep video prediction model conditioned on a single image and an action class. To generate future frames, we first detect keypoints of a moving object and predict future motion as a sequence of keypoints. The input image is then translated following the predicted keypoints sequence to compose future frames. Detecting the keypoints is central to our algorithm, and our method is trained to detect the keypoints of arbitrary objects in an unsupervised manner. Moreover, the detected keypoints of the original videos are used as pseudo-labels to learn the motion of objects. Experimental results show that our method is successfully applied to various datasets without the cost of labeling keypoints in videos. The detected keypoints are similar to human-annotated labels, and prediction results are more realistic compared to the previous methods.

## 1  Introduction

Video prediction is a task of synthesizing future video frames from a single or few image(s), which is challenging due to the uncertainty of the dynamic motions in scenes. Despite its difficulty, this task has attracted great interests in machine learning, as predicting unknown future is fundamental to understanding video data and the physical world.

Early works in video prediction adopted deterministic models that directly minimize the pixel distance between the generated frames and ground-truth frames [1–4]. Srivastava *et. al.* [1] studied LSTM-based model for video prediction and video reconstruction. Finn *et. al.* [2] generate the next frame by pixel-wise transformation on the previous frame. Kalchbrenner *et. al.* [3] generate a future frame by calculating the distribution of RGB values per pixel given prior frames. De Brabandere *et. al.* [4] propose a model that generates dynamic convolutional filters for video and stereo prediction. These deterministic models tend to produce blurry results, and also have a fundamental limitation in that they have difficulty in generating videos for novel scenes that they have not seen before. To overcome these issues, recent approaches take on generative methods based on generative adversarial networks (GANs) [5] and variational auto-encoders (VAEs) [6], by using the adversarial loss of GANs and the KL-divergence loss of VAEs as an additional training loss [7–10]. Babaeizadeh *et. al.* [7] extend the work of Finn *et. al.* [2] by using the VAE as a backbone structure to generate various samples. Mathieu *et. al.* [8] propose a GAN based model to handle blurry results induced by MSE loss. Lee *et. al.* [9] introduced a model combining both GAN and VAE to generate sharp and various results. Denton *et. al.* [10] aim to generate various results by learning the conditional distribution of latent variables that drive the next frame with the VAE as a backbone structure.

Aforementioned works fall into a black-box approach in Fig. 1 (a), where videos are directly synthesized through spatio-temporal networks. This type of approach achieved limited success on few simple datasets which have low variance such as the Moving MNIST [1], KTH human actions [11], and BAIR action-free robot pushing dataset [12].

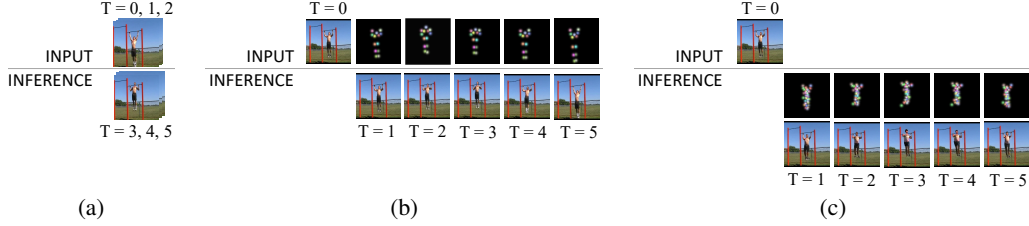

Figure 1: Different types of video prediction algorithms. (a) predicts video using spatio-temporal models with a black-box approach. (b) utilizes human-annotated keypoints labels and uses it as a guidance for future frames generation. (c) is our proposed method that internally generates keypoints labels by training the keypoints detector in an unsupervised manner. It also guides future frames with the keypoints sequence.

As one can imagine, it is more difficult to generate videos than images as we need to represent the temporal domain in addition to the spatial domain. To make the model to predict future with the comprehension of this nature of videos, some works have attempted to train the model by disentangling the spatial (contents) and the temporal (motion) characteristics of videos [13–16]. Tulyakov *et. al.* [13] proposed to generate videos with two random values, each representing the contents and the motion feature of the video. The method of Villegas *et. al.* [14] predicts the next frame with latent motion feature related to multiple previous difference images. To better decompose two features, the works of [15, 16] impose adversarial loss on each feature. However, the results of these works are similar to the deterministic methods in quality.

Meanwhile, recent image translation works have shown that using keypoints is a promising approach [17–20]. In these works, keypoints are used as a guidance for the image translation leading to qualitative improvements of the results. This approach was extended for the video prediction task by [21–23], which fall into Fig. 1 (b). These methods generate future frames by translating a reference image using the keypoints sequence as a guidance. The works in [21, 22] are prediction models that utilize labels of human joint positions. Villegas *et. al.* [21] succeeded in generating long-term future image sequence and improving visual quality of the results by applying a method called visual-structure analogy making based on the work of Reed *et. al.* [24]. Cai *et. al.* [22] proposed an integrated model that is capable of video generation, prediction and completion task by optimizing latent variables in accordance with given constraints. Wang *et. al.* [23] employed the VAE network to generate diverse samples and use a keypoints sequence for synthesizing a human face image sequence. These works suggest that using keypoints is effective for the video prediction task. They all produce high-quality results for natural scene datasets such as the Penn Action [25] and UCF-101 [26]. However, these works require frame-by-frame keypoints labeling, which limits the applicability of the methods.

A way to deal with this problem is to employ a keypoints detector trained in an unsupervised manner. Several models of this kind have recently been proposed [27–29]. The method of Thewlis *et. al.* [27] learns to detect keypoints using a known transformation function between two images. Zhang *et. al.* [28] proposed to find keypoints for image reconstruction and manipulation tasks. This model is based on the VAE with the hourglass network [30], and imposes constraints on detected landmarks to enhance the validity of the results. Jakab *et. al.* [29] proposed an unsupervised approach to find keypoints of an object that serve as the guidance in image translation task. The work uses a simple method called the heatmap bottleneck, showing the state-of-the-art keypoints detection performance without imposing any regularization. This type of keypoints detector was also studied for a video generation model that implants the motion of a source video to a static object of the target image by Siarohin *et. al.* [31], showing successful results on various video datasets.

Building on ideas from previous works, we propose a deep video prediction model that includes a keypoints detector trained in an unsupervised manner which is illustrated in Fig. 1 (c). Compared with Fig. 1 (a) and (b), our model performs better on various datasets including the datasets without the ground-truth keypoints labels, as our method learns the keypoints best suited for the video synthesis without labels. Fig. 2 shows the overview of our method for predicting future frames at the inference time. Our approach consists of 3 stages: keypoints detection, motion generation, and keypoints-guided image translation. In our work, no labels except for the action class are demanded. Given an input image and a target action class, our method first predicts the keypoints of the input

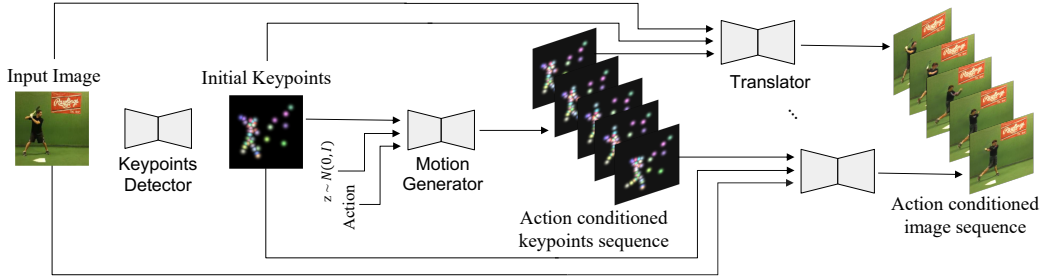

Figure 2: The overview of our method at inference time. Our method generates future frames through 3 stages: keypoints detection, motion generation, and keypoints-guided image translation.

image. Then, our method generates a sequence of keypoints starting from the predicted keypoints, which follows the given action. Finally, the output video is synthesized by translating the input image frame-by-frame using the generated keypoints sequence as a guidance. The key in our unsupervised approach is to use the keypoints of ground-truth videos detected from the keypoints detector as pseudo-labels for learning the motion generator. Moreover, we propose a robust image translator using the analogical relationship between the image and keypoints, and a background masking to suppress the distraction from noisy backgrounds. Experimental results show that our method produces better results than previous works, even the ones that utilizes human-annotated keypoints labels. The performance of the keypoints detector is greatly improved allowing our method to be applied to various datasets.

The summary of our contributions is as follows.

- We propose a deep generative method for class-conditional video prediction from a single image. Our method internally generates keypoints of the foreground object to guide the synthesis of future motion.

- Our method learns to generate a variety of keypoints sequences from data without labels, which enables our method to model the motion of arbitrary objects including human, animal, and etc.

- Our method is robust to the noise of data such as distracting backgrounds, allowing our method to work robustly on challenging datasets.

## 2 Method

Given a source image $\mathbf{v}_0 \in \mathbb{R}^{H \times W \times 3}$ at $t = 0$ with a target action vector $\mathbf{a} \in \mathbb{R}^C$, the goal of our task is to predict future frames $\hat{\mathbf{v}}_{1:T} \in \mathbb{R}^{T \times H \times W \times 3}$ with $T > 0$, where the motion of a foreground object follows the action code. To tackle this problem, our approach is to train a deep generative network, consisting of a keypoints detector, a motion generator, and a keypoints-guided image translator. Instead of generating $\hat{\mathbf{v}}_{1:T}$ at once, we first predict future motion of the object as a keypoints sequence $\hat{\mathbf{k}}_{1:T} \in \mathbb{R}^{T \times K \times 2}$ and translate the input frame $\mathbf{v}_0$ with $\hat{\mathbf{k}}_{1:T}$ as a guidance. The training process consists of two stages: (i) learning the keypoints detector with the image translator and (ii) learning the motion generator. In the following, we describe our network and its training method in detail.

**Learning the keypoints detector with the image translator.** Fig. 3 shows our method for the image translation employing the keypoints detector and the keypoints-guided image translator. Inspired by [29], our method learns to detect the keypoints of a foreground object by learning the image translation between two frames $(\mathbf{v}, \mathbf{v}')$ in the same video. The intuition behind learning the keypoints in this way is that translating $\mathbf{v}$ close to $\mathbf{v}'$ enforces the network to automatically find the most dynamic parts of the image, which can then be used as the guidance to move the object in the reference image. Different from [29], the target image is synthesized by inferring the analogical relationship [21, 24] between the keypoints and the image, where the difference between the reference and the target image $(\mathbf{v}, \mathbf{v}')$ corresponds to the difference between the two detected keypoints sets, $(\hat{\mathbf{k}}, \hat{\mathbf{k}}')$.

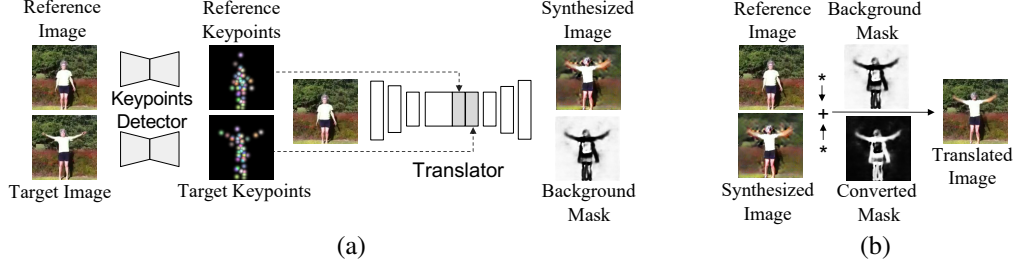

(a)                                    (b)

Figure 3: The overview of training the keypoints detector and the image translator. (a) shows the the unsupervised learning of the keypoints by learning the image translation. (b) shows the detail of our background masking method.

The keypoints detector $Q$ finds $K$ keypoints of the input image. The keypoints coordinates $\hat{\mathbf{k}} \in \mathbb{R}^{K \times 2}$ are obtained by calculating the expected coordinates of the $K$-channel soft binary map $\mathbf{l} \in \mathbb{R}^{H \times W \times K}$, which is the last feature map of $Q$ followed by a softmax activation described as

$$
\begin{aligned}
\mathbf{l}^n &= \frac{e^{Q(\mathbf{v})^n}}{\sum_{\mathbf{u}} e^{Q(\mathbf{v})_{\mathbf{u}}^n}}, \\
\hat{\mathbf{k}}^n &= \sum_{\mathbf{u}} \mathbf{u} \cdot \mathbf{l}_{\mathbf{u}}^n,
\end{aligned}
\tag{1}
$$

where $\hat{\mathbf{k}}^n$ is the coordinates of the $n$-th keypoint and $\mathbf{u}$ is the pixel coordinates. The detected keypoints $\hat{\mathbf{k}}$ are then normalized to have values between -1 and 1, and converted to $K$ gaussian distribution maps $\mathbf{d} \in \mathbb{R}^{h \times w \times K}$ using the following formulation:

$$
\hat{\mathbf{d}}_{\mathbf{u}'}^n = \frac{1}{\sigma \sqrt{2\pi}} e^{-\left(\mathbf{u}' - \hat{\mathbf{k}}^n\right)^2 / 2\sigma^2},
\tag{2}
$$

where $\sigma$ is the standard deviation of a Gaussian distribution.

Our image translation network $T$ only handles dynamic regions by generating image $\mathbf{s} \in \mathbb{R}^{H \times W \times 3}$ with a new appearance of the object and a soft background mask $\mathbf{m} \in \mathbb{R}^{H \times W \times 1}$ similar to [32]. Then, we smoothly blend the input image $\mathbf{v}$ and synthesized image $\mathbf{s}$ using the background mask $\mathbf{m}$ which is described as

$$
\begin{aligned}
\mathbf{m}, \mathbf{s} &= T(\mathbf{v}; \hat{\mathbf{k}}; \hat{\mathbf{k}}') \\
\hat{\mathbf{v}} &= \mathbf{m} \odot \mathbf{v} + (1 - \mathbf{m}) \odot \mathbf{s},
\end{aligned}
\tag{3}
$$

where $\odot$ refers to a Hadamard product of two tensors.

The training objective for $Q$ and $T$ consists of a reconstruction loss defined by the distance between the output and target image, and an adversarial loss [5] that leads our model to produce realistic images. We use the perceptual loss [33] based on the VGG-19 network [34] pretrained for image recognition task [35] as the reconstruction loss to enforce perceptual similarity of the generated image and the target image.

Hence, our optimization is to alternately minimize the two losses defined as follows:

$$
\begin{aligned}
L_{D_{im}} &= -\log D_{im}(\mathbf{v}') - \log(1 - D_{im}(\hat{\mathbf{v}})) \\
L_{Q,T} &= -\log D_{im}(\hat{\mathbf{v}}) + \lambda_1 \mathbb{E}_l \|\Phi_l(\hat{\mathbf{v}}) - \Phi_l(\mathbf{v}')\|,
\end{aligned}
\tag{4}
$$

where $D_{im}$ is the image discriminator, $\Phi_l$ is the $l$-th layer of the VGG-19 network, and $\lambda_1$ is the weight of the perceptual loss.

**Learning the motion generator with pseudo-labeled data.** Our method for the motion generation is shown in Fig. 4. After completing the first training stage, we can detect keypoints of any image and translate image with arbitrary target keypoints. With the trained keypoints detector, we prepare pseudo-labels $\hat{\mathbf{k}}_{0:T}$ by detecting keypoints of real videos and use them to train our motion generator $M$ to generate sequences of future keypoints, which is used as a guidance for synthesizing future frames $\hat{\mathbf{v}}_{1:T}$.

We build our motion generator upon a conditional variational auto-encoder (cVAE) [36] to learn the distribution of future events with the given conditions. Specifically, our motion generator learns to

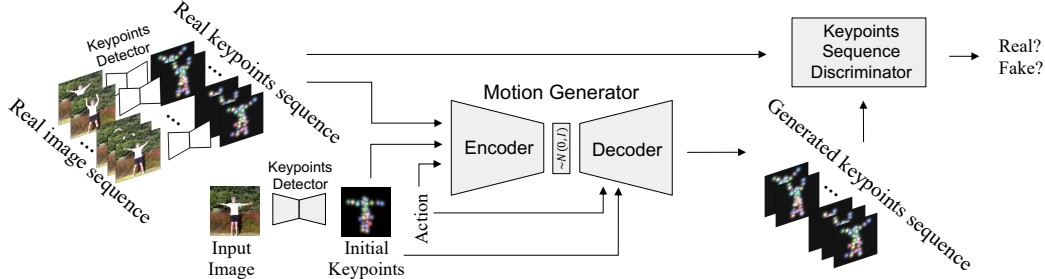

Figure 4: The overview of training the motion generator. Our motion generator is built upon cVAE framework conditioned on the initial keypoints and the action class. We utilize detected keypoints sequence of real videos to learn the motion of arbitrary objects.

encode the pseudo-labels to normally distributed latent variables $q_\phi(\mathbf{z}|\hat{\mathbf{k}}_{1:T}, \hat{\mathbf{k}}_0, \mathbf{a})$, and to decode $\mathbf{z}$ back to the corresponding keypoints sequence $p_\theta(\hat{\mathbf{k}}_{1:T}|\mathbf{z}, \hat{\mathbf{k}}_0, \mathbf{a})$. To handle the sequential data, an LSTM network [37] was used for both the encoder and the decoder. At inference time, the future motion is predicted by random sampling of $\mathbf{z}$ value from $\mathcal{N}(0, I)$, which leads $M$ to generate many possible results.

The network is trained by optimizing the variational lower bound [6] that is comprised of the KL-divergence and the reconstruction loss. We additionally trained the keypoints sequence discriminator $D_{seq}$, as we have found that using an adversarial loss [5] on the cVAE model improves the quality of the results.

Our training of $M$ is to alternately minimize the following two objectives:

$$
\begin{aligned}
L_{D_{seq}} &= -\log D_{seq}(\hat{\mathbf{k}}_{1:T}) - \log(1 - D_{seq}(\tilde{\mathbf{k}}_{1:T})) \\
L_M &= D_{KL}(q_\phi(\mathbf{z}|\hat{\mathbf{k}}_{1:T}; \hat{\mathbf{k}}_0; \mathbf{a}) \| p_z(\mathbf{z})) + \lambda_2 \|\tilde{\mathbf{k}}_{1:T} - \hat{\mathbf{k}}_{1:T}\|_1 - \lambda_3 \log D_{seq}(\tilde{\mathbf{k}}_{1:T}),
\end{aligned}
\tag{5}
$$

where $\tilde{\mathbf{k}}$ is the reconstruction of keypoints, and $\lambda_2$ and $\lambda_3$ are hyperparameters. The prior distribution of latent variables $p_z(\mathbf{z})$ is set as $\mathcal{N}(0, I)$.

## 3 Experiments

### 3.1 Datasets

**Penn Action**  This dataset [25] consists of videos of human in sports action. The total number of videos is 2326, and the number of action class is 15. We only used videos that show the whole body of a foreground actor and excluded classes with too few samples. Hence, out of 15 action classes, we only used 9 classes – baseball pitch, clean and jerk, pull ups, baseball swing, golf swing, tennis forehand, jumping jacks, tennis serve, and squats. Due to the lack of data, only 10 samples per each class were used as the test set and the rest as the training set, making sure that there are no overlapping scenes in the training and test sets. The final dataset consists of 1172 training videos and 90 test videos. During the training process, data was intensely augmented by random horizontal flipping, random rotation, random image filter, and random cropping.

**UvA-NEMO and MGIF**  The UvA-NEMO [38] consists of 1234 videos of smiling human faces, which is split into 1110 videos for the training set and 124 for the evaluation set. The MGIF [31] is a dataset consisting of videos of cartoon animal characters simply walking or running on a white-colored background. For this dataset, 900 videos are used for the training and 100 videos are used for the evaluation. We used the pre-processed version of both datasets provided by Siarohin *et. al.* [31], and applied the same augmentation methods used for the Penn Action dataset.

### 3.2 Implementation details

The resolution of both the input and the output images is $128{\times}128$, and the number of keypoints $K$ was set to 40, 15, and 60 for the Penn Action, UvA-NEMO, and MGIF dataset, respectively. We implemented our method using TensorFlow with the Adam optimizer [41], the learning rate of 0.0001,

| Dataset | [39] | [16] | [21] | Ours |
|---|---|---|---|---|
| Penn Action [25] | 4083.3 | 3324.9 | 2187.5 | **1509.0** |
| UvA-NEMO [38] | 666.9 | 265.2 | - | **162.4** |
| MGIF [31] | 683.1 | 1079.6 | - | **409.1** |

Table 1: Fréchet Video Distance (FVD) [40] of generated videos. On every datasets, our method achieved the best score. (The lower is better.)

the batch size of 32, and the two momentum values of 0.5 and 0.999. We decreased the learning rate by 0.95 for every 20,000 iterations. The keypoints detector and the translator were optimized until the convergence of the perceptual loss, and the motion generator until the KL-divergence convergence. Considering the tendency of the convergence, $\lambda_1$, $\lambda_2$, and $\lambda_3$ were set to 1, 1000, and 2, respectively. Since the UvA-NEMO and MGIF datasets consist of videos of same action, only initial keypoints $\mathbf{k}_0$ are set as the condition for the motion generation. [1]

### 3.3 Baselines

We compare our method with three baselines [16, 39, 21], all of which produce future frames in two stages predicting the guiding information first. The method of Wichers *et. al.* [16] generates frames with the latent motion feature, which is learned with an adversarial network. The works of Villegas *et. al.* [21] and Li *et. al.* [39] respectively guide frame generation with keypoints and optical flow. Each model utilizes keypoints labels and pretrained optical flow predictor. Only the work of Li *et. al.* [39] is conditioned on a single frame like ours, while others [16, 21] are conditioned on multiple prior frames. For implementing the baseline models, we used the codes released by the authors maintaining original settings of each model, including the number of conditional images, image resolution and, the number of future frames.

### 3.4 Results

**Qualitative results**    Video prediction results of our method on Penn Action dataset is shown in Fig. 5 (a). The generated videos present both the realistic image per frame and the plausible motion corresponding to the target action class. The synthesized image and mask sequence imply that our model disentangles dynamic regions well, and the predicted keypoints sequence is similar to human-annotated labels. Comparison of the results are shown in Fig. 5 (b). Since the number of generated frames varies from model to model, we sampled 8 frames from each result that represent the whole sequence for the qualitative comparison. The results imply that our method achieved improvements in both the visual and the dynamics quality compared to the baselines. The work of Wichers *et. al.* [16] failed to make realistic and dynamic future frames, although it is capable of distinguishing moving objects to some extent. The method [39] struggles with the error propagation since they apply the warp operation with the predicted optical flow sequence. The results of Villegas *et. al.* [21] are the best among the baselines, showing plausible and dynamic motion. However, the results of our model are more visually realistic, as it employs the keypoints specifically learned for the image synthesis. Moreover, our model can generate various results with only one image as shown in Fig. 6, by randomly sampling the **z** value and changing the target action class.

Fig. 7 shows our prediction results on the UvA-NEMO and the MGIF datasets. The work of [21] was not compared since these datasets have no keypoints labels. The results show that the work of Li *et. al.* [39] still has the error propagation issue on both datasets due to the warping operation. The method of Wichers *et. al.* [16] failed on the MGIF dataset, but succeeded at generating plausible future frames on the UvA-NEMO dataset. Meanwhile, our method generates frames with dynamic motion while maintaining the visual attributes of the foreground object over the whole sequence.

In addition to the video prediction results, we also demonstrate the performance of our image translator in Fig. 8. Examples are chosen to show different capabilities of our image translator: (a) translation, (b) inpainting, and (c) object removal. The result in (a) suggests that the reference image is well translated by detected keypoints. The synthesized mask and the image imply that our model focuses on filling in occluded or disoccluded regions, separating the foreground region

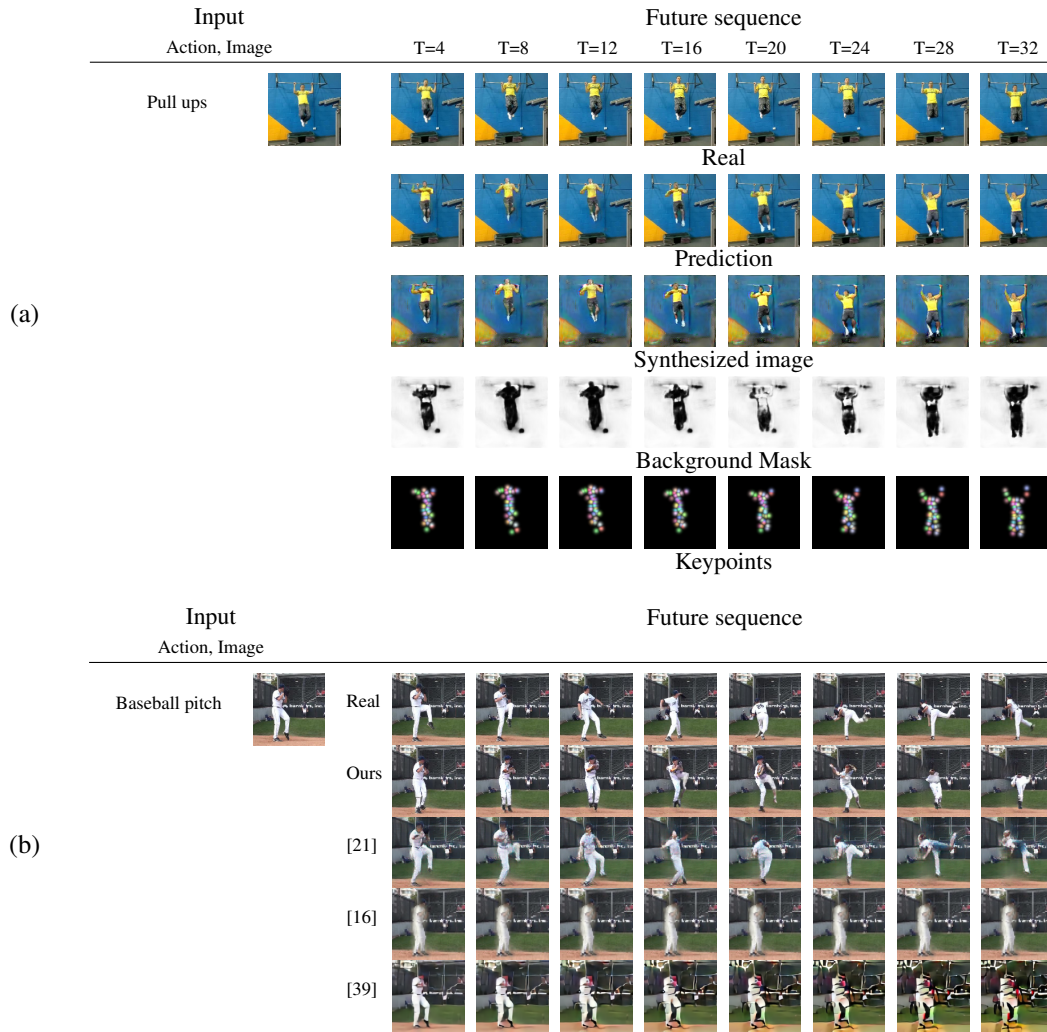

Figure 5: Video prediction results on the Penn Action dataset. In (a), the input image and the target action are shown on the left side. On the right side, the ground-truth video, synthesized video after masking, synthesized video before masking, background mask, and keypoints are shown from the top. (b) compares the result of ours with the baseline methods.

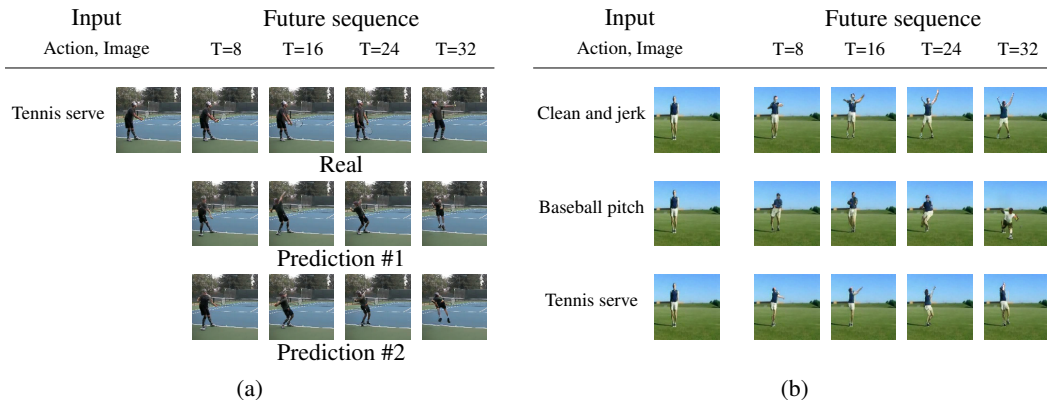

Figure 6: Variety of prediction results. The examples in (a) are induced by the random sampling of $\mathbf{z}$ value in the motion generator, and (b) by the change of the target action class $\mathbf{a}$.

Input          Future sequence          Input          Future sequence

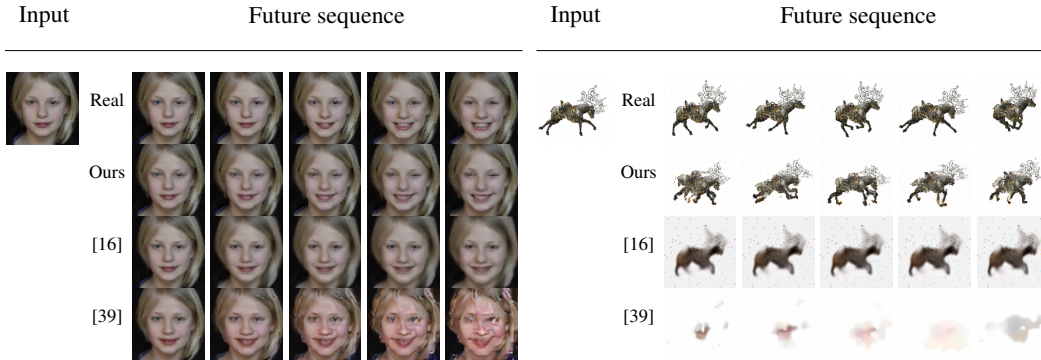

Figure 7: Video prediction results on UvA-NEMO and MGIF datasets.

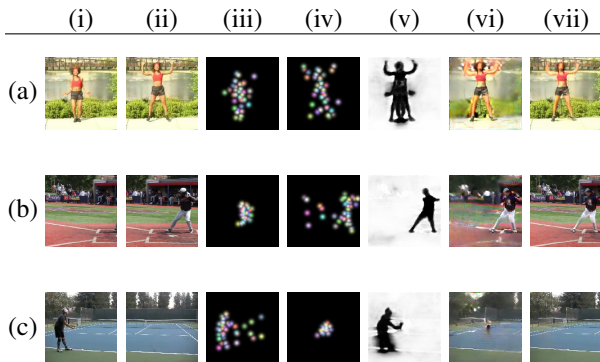

Figure 8: Image translation results. Columns represent the following in order – reference image, target image, detected keypoints of reference/target images, background mask, synthesized image, and final translation result. The samples in rows (a)-(c) show that our image translator is capable of different tasks including translation, inpainting, and object removal.

| Method | Accuracy |
|---|---|
| Ours | **68.89** |
| Ours w/o **a** | 63.33 |
| [21] | 47.14 |
| [16] | 40.00 |
| [39] | 15.55 |

Table 2: Action recognition accuracy.

| Method | Ranking |
|---|---|
| Ours | **1.81** $\pm$ 1.02 |
| [21] | 2.44 $\pm$ 0.98 |
| [16] | 3.14 $\pm$ 1.09 |
| [39] | 2.61 $\pm$ 0.96 |

Table 3: Quantitative result of the user study. The values refer to average rankings.

sharply. Interestingly, the results in (b) and (c) show that our model learned additional abilities to fill or remove parts of the image, when there are no corresponding objects. All these imply that our model has learned the robust ability to discern moving objects as keypoints.

**Quantitative results**   For quantitative comparison, we used the Fréchet video distance (FVD) [40]. This is Fréchet distance between the feature representations of real and generated videos. The feature representations were gained from the I3D model [42] trained on kinetic-400 [43]. The results are reported in Table 1. Our method achieved the smallest FVD values on every datasets. This implies that our method generates more realistic videos compared to the baseline methods.

In addition, we assess the plausibility of generated motion. Since it is obvious what action the object would take from the conditional image(s), we compared the action recognition accuracy on the results using the two-stream CNN [44] that is fine-tuned on the Penn Action dataset. We additionally compared the results of our method without the conditional term for the target action class for fair comparison. Our method achieved the best recognition score as shown in Table 2. Even though removing the action class condition slightly affects the performance, the gap is small compared to the baseline results. This implies that our method does a better job of generating plausible motion compared to the other baseline approaches.

We also conducted a user study on Amazon Mechanical Turk (AMT), since above methods cannot fully reflect the human perception on the visual quality of the results. We compared 70 of the 90 prediction results on the Penn Action dataset, since [21] was trained for different set of action classes.

| | Input | | | Output |
|---|---|---|---|---|
| | v | v′ | k | k′ | v̂ |

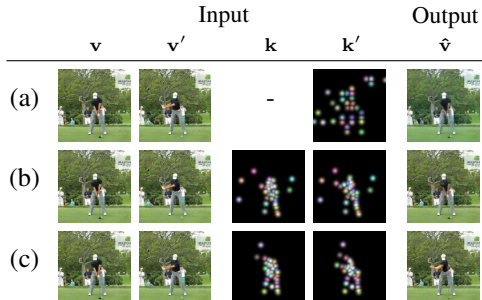

Figure 9: The results of the keypoints-guided image translation from (a) the baseline method [29], (b) our network without the mask, and (c) our network. Our method achieved performance improvement in both the keypoints detection and the image translation compared to the baseline method.

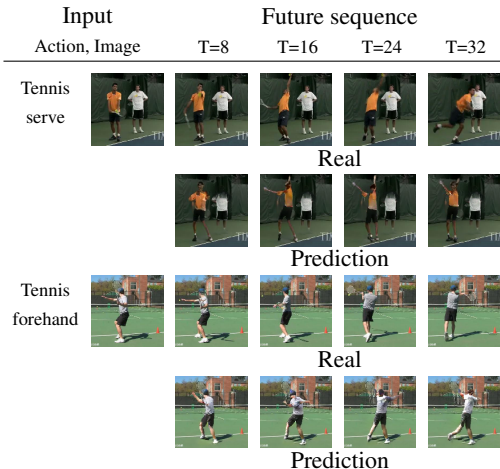

| Input | Future sequence | | | |
|---|---|---|---|---|
| Action, Image | T=8 | T=16 | T=24 | T=32 |

Figure 10: Failure cases.

15 workers were asked to rank the results generated by different methods based on the visual quality and the degree of the movement in foreground region. During the process, workers were shown four videos side by side, where the order of videos was randomly chosen for each vote. The averaged rankings for all methods are shown in Table 3, which indicates that our method outperforms all the baselines in both aspects, even though it has been trained without any labels.

**Component analysis** Our keypoints-guided image translation method achieved improvement in performance compared to the original work [29] by (i) learning the analogical relationship between the keypoints and the image and (ii) generating a background mask. We analyzed the effect of each component as shown in Fig. 9.

Comparing the results in Fig. 9 (a) and (b), the performance of the keypoints detector has improved when the process employs the reference keypoints in addition to the target keypoints. With keypoints in both the images and the reference image, the translator can synthesize the foreground object in the target pose by inferring the analogical relationship like "A is to B as C is to what?". If the reference keypoints are not considered, the translator would have to find the region to translate independently which is redundant and inefficient set-up. The result in Fig. 9 (c) shows that incorporating the background mask generation into the keypoints-guided image translation led to significant improvement in quality of the translated image. The mask generation is effective when only a specific part of the image needs to be translated, since synthesizing only the foreground object is beneficial for the network to fool the image discriminator by reducing the complexity of the modeling compared to synthesizing the entire scene. Achieving these improvements in the keypoints detection and the image translation, our method could be applied to various datasets successfully.

**Failure Cases** We found two cases in which our model failed to generate plausible future frames. The first case is from the failure of the keypoints detector when there are multiple objects with similar size in the input image (first example in Fig. 10). This failure causes a series of failures in the motion generation and the image translation. The second example in Fig. 10 shows the other failure case. Since our keypoints detector works in a body orientation agnostic way, object moves in the opposite direction from our expectations in some cases.

## 4 Conclusion

In this paper, we proposed an action-conditioned video prediction method using a single image. Instead of generating future frames at once, we first predict the temporal propagation of the foreground object as a sequence of keypoints. Following the motion of the keypoints, input image is translated to compose future frames. Our network is trained in an unsupervised manner using the predicted keypoints of the original videos as pseudo-labels to train the motion generator. Experimental results show that our method achieved significant improvement on visual quality of the results and is successfully applied to various datasets without using any labels.

**Acknowledgement** This work was supported by Samsung Research Funding Center of Samsung Electronics under Project Number SRFC-IT1701-01.

## Footnotes

[1]The architectural details of our model are demonstrated in the supplementary material.

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
