[Supplementary Material · Unsupervised_Keypoint_Learning_for_Guiding_Class-Conditional_Video_Prediction_Supp.pdf]

# (Supplementary Material)
# Unsupervised Keypoint Learning
# for Guiding Class-Conditional Video Prediction

**Yunji Kim[1], Seonghyeon Nam[1], In Cho[1], and Seon Joo Kim[1,2]**
[1]Yonsei University　　　　[2]Facebook
{kim_yunji,shnnam,join,seonjookim}@yonsei.ac.kr

## 1 Network Architecture

The network designs for the keypoints detector, the keypoints-guided image translator, and the image discriminator are detailed in Table 1, and the details for the motion generator and the keypoints sequence discriminator are in Table 2.

## 2 Training

The training process consists of two stages. We first train the keypoints detector and keypoints-guided image translator with the image discriminator. Then the motion generator with the keypoints sequence discriminator is trained. Details of each stage are described in Algorithm 1.

## 3 Visualization of the Guiding Information

To analyze the effectiveness of the guiding information, we visualized the information along with the final results in Fig. 1. The results show that methods utilizing keypoints perform better than others, generating visually pleasing video while discerning the foreground objects well. Another advantage of the keypoints representation is being able to visualize the motion clearly.

## 4 Additional Experimental Results

Fig. 2 and Fig. 3 show more prediction results on Penn Action dataset. The examples in Fig. 4 and Fig. 5 show that our method can generate various results from the identical image. Fig. 6 and Fig. 7 present more prediction results on UvA-NEMO and MGIF datasets.

## References

[1] R. Villegas, J. Yang, Y. Zou, S. Sohn, X. Lin, and H. Lee, "Learning to generate long-term future via hierarchical prediction," in *ICML*, 2017.

[2] N. Wichers, R. Villegas, D. Erhan, and H. Lee, "Hierarchical long-term video prediction without supervision," in *ICML*, 2018.

[3] Y. Li, C. Fang, J. Yang, Z. Wang, X. Lu, and M. Yang, "Flow-grounded spatial-temporal video prediction from still images," in *ECCV*, 2018.

| Module | | Layers | Input size | Output size |
|---|---|---|---|---|
| Keypoints Detector (U-Net) | Encoder | CONV(7,1), CONV(3,1) | $128{\times}128{\times}3$ | $128{\times}128{\times}32$ |
| | | CONV(3,2), CONV(3,1) | $128{\times}128{\times}32$ | $64{\times}64{\times}64$ |
| | | CONV(3,2), CONV(3,1) | $64{\times}64{\times}64$ | $32{\times}32{\times}128$ |
| | | CONV(3,2), CONV(3,1) | $32{\times}32{\times}128$ | $16{\times}16{\times}256$ |
| | Decoder | CONV(3,1)$\times$2, Upsample(2) | $16{\times}16{\times}256$ | $32{\times}32{\times}128$ |
| | | CONV(3,1)$\times$2, Upsample(2) | $32{\times}32{\times}(128{+}128)$ | $64{\times}64{\times}64$ |
| | | CONV(3,1)$\times$2, Upsample(2) | $64{\times}64{\times}(64{+}64)$ | $128{\times}128{\times}32$ |
| | | CONV(3,1)$\times$2 | $128{\times}128{\times}(32{+}32)$ | $128{\times}128{\times}16$ |
| | | Conv2d(1,1) | $128{\times}128{\times}16$ | $128{\times}128{\times}K$ |
| | | Spatial-wise Softmax function | $128{\times}128{\times}K$ | (a)$128{\times}128{\times}K$ |
| Translator | Encoder | CONV(7,1) | $128{\times}128{\times}3$ | $128{\times}128{\times}32$ |
| | | CONV(3,1) | $128{\times}128{\times}32$ | $128{\times}128{\times}32$ |
| | | [CONV(3,2), CONV(3,1)]$\times$2 | $128{\times}128{\times}32$ | $32{\times}32{\times}128$ |
| | Decoder | CONV(3,1)$\times$4 | (b)$32{\times}32{\times}(128{+}K{+}K)$ | $32{\times}32{\times}256$ |
| | | Upsample(2) | $32{\times}32{\times}256$ | $64{\times}64{\times}256$ |
| | | CONV(3,1)$\times$4 | $64{\times}64{\times}256$ | $64{\times}64{\times}128$ |
| | | Upsample(2) | $64{\times}64{\times}128$ | (c)$128{\times}128{\times}128$ |
| | | Conv2d(3,1) | (c) $128{\times}128{\times}128$ | (d)$128{\times}128{\times}3$ |
| | | Conv2d(3,1), Sigmoid | (c) $128{\times}128{\times}128$ | (e)$128{\times}128{\times}1$ |
| Image Dicriminator | | [Conv2d(4,2), LeakyReLU(0.01)]$\times$6 | $128{\times}128{\times}3$ | $4{\times}4{\times}2048$ |
| | | Conv2d(3,1,padding=1) | $4{\times}4{\times}2048$ | $6{\times}6{\times}1$ |

Table 1: The network designs for the keypoints detector, the keypoints-guided image translator, and the image discriminator. "CONV(k,s)" represents the sequence of layers, [Conv2d(k,s), BatchNorm, RELU]. Parameters k and s are for the kernel and the stride size of the convolutional layer. "Upsample(x)" resizes the map with a ratio of x through the bilinear interpolation. (a) is $\mathbf{l}$ in the Equation 1 of the main paper. In the translator, keypoints are converted to $K$-depth gaussian distribution maps each centered at keypoints coordinates. (b) represents the size of the concatenation of image feature map of the reference image ($\mathbf{v}$) and two gaussian distribution maps ($\mathbf{d}, \mathbf{d}'$). (d) and (e) are $\mathbf{s}$ and $\mathbf{m}$ in the Equation 3 of the main paper, which are generated by layers branched off at (c).

| Module | | Layers | Input size | Output size |
|---|---|---|---|---|
| Motion Generator | Encoder | Multiple-LSTM(1024,1024) | (a)$32{\times}(K{\times}2)$ | 1024 |
| | | Linear | (b)$1024{+}(K{\times}2){+}C$ | (c) $\mathbf{z}$ |
| | Decoder | Linear | $\mathbf{z}{+}(K{\times}2){+}C$ | 32 |
| | | Multiple-LSTM(1024,1024) | 32 | (d)$32{\times}(K{\times}2)$ |
| Sequence Discriminator | | Multiple-LSTM(1024,1024) | $32{\times}(K{\times}2)$ | 1 |

Table 2: The network designs for the motion generator and the keypoints sequence discriminator. "Multiple-LSTM($c_1, ..., c_n$)" stands for the stacked lstm layers with the feature dimension of each layer as $c_n$. (a) represents the size of the pseudo-label, which is the coordinates sequence of $K$ keypoints with length of 32. (b) represents the concatenation of the keypoints sequence embedding vector, coordinates of the initial keypoints and 1-hot representation of the target action class. (c) is the mean and variance of the latent variables mapped by the cVAE encoder. (d) is the size of the predicted keypoints sequence $\tilde{\mathbf{k}}_{1:T}$, which is the future motion of a foreground object.

---

**Algorithm 1** Training Process

---

**I. Train the keypoints detector($Q$), translator($T$) and image discriminator($D_{im}$)**

   Set the learning rates $\alpha_1, \alpha_2, \alpha_3$ and initialize the network parameters $\theta_{D_{im}}, \theta_Q, \theta_T$
   Until $\mathbb{E}_l \|\Phi_l(\hat{\mathbf{v}}) - \Phi_l(\mathbf{v}')\|$ converges:

       Randomly generate the batch of pair of images, $S_{(\mathbf{v},\mathbf{v}')}$
       Detect keypoints of two images ; $\hat{\mathbf{k}} = Q(\mathbf{v}), \hat{\mathbf{k}}' = Q(\mathbf{v}')$
       Translate the reference image according to target keypoints ; $\hat{\mathbf{v}} = T(\mathbf{v}; \hat{\mathbf{k}}; \hat{\mathbf{k}}')$
       Update parameters ; $\theta_{D_{im}} \mathrel{-}= \alpha_1 \nabla_{\theta_{D_{im}}} \mathbb{E}_{(\hat{\mathbf{v}},\mathbf{v}')} L_{D_{im}},$
   $$\theta_Q \mathrel{-}= \alpha_2 \nabla_{\theta_Q} \mathbb{E}_{(\hat{\mathbf{v}},\mathbf{v}')} L_{Q,T},$$
   $$\theta_T \mathrel{-}= \alpha_3 \nabla_{\theta_T} \mathbb{E}_{(\hat{\mathbf{v}},\mathbf{v}')} L_{Q,T}$$

**II. Train the motion generator($M$) and keypoints sequence discriminator ($D_{seq}$)**

   Make pseudo-labels $\{\hat{\mathbf{k}}^0_{0:T}, \hat{\mathbf{k}}^1_{0:T}, ...\}$ using the trained keypoints detector
   Set the learning rates $\alpha_1, \alpha_2$ and initialize the network parameters $\theta_{D_{seq}}, \theta_M$
   Until $D_{KL}(q_\phi(\mathbf{z}|\hat{\mathbf{k}}_{1:T}; \hat{\mathbf{k}}_0; \mathbf{a}) \| p_z(\mathbf{z}))$ converges:

       Randomly generate the batch of keypoints sequence $S_{\hat{\mathbf{k}}_{0:T}}$
       Encode keypoints sequence $\hat{\mathbf{k}}_{1:T}$ into the latent variables $\mathbf{z}$
       Generate keypoints sequence $\tilde{\mathbf{k}}_{1:T}$ using the encoded latent values of $\mathbf{z}$
       Update parameters ; $\theta_{D_{seq}} \mathrel{-}= \alpha_1 \nabla_{\theta_{D_{seq}}} \mathbb{E}_{(\tilde{\mathbf{k}}_{1:T}, \hat{\mathbf{k}}_{0:T})} L_{D_{seq}},$
   $$\theta_M \mathrel{-}= \alpha_2 \nabla_{\theta_M} \mathbb{E}_{(\tilde{\mathbf{k}}_{1:T}, \hat{\mathbf{k}}_{0:T})} L_M$$

---

Figure 1: Visualization of the guiding information along with the final results. The keypoints of our model and [1], background mask of [2] and optical flow of [3]. The keypoints of our model is learned in an unsupervised manner, while those of [1] are trained with human-annotated labels.

Figure 2: Video prediction results on Penn Action dataset.

Figure 3: Comparison of video prediction results on Penn Action dataset.

| Input | Future sequence | | | | | | | |
|---|---|---|---|---|---|---|---|---|
| Action, Image | T=4 | T=8 | T=12 | T=16 | T=20 | T=24 | T=28 | T=32 |

Tennis serve

Jumping jacks

Baseball swing

Baseball swing

Squat

Baseball pitch

Figure 4: Variety of prediction results induced by the change of the target action class **a**.

| Input | Future sequence | | | | | | | |
|---|---|---|---|---|---|---|---|---|
| Action, Image | T=4 | T=8 | T=12 | T=16 | T=20 | T=24 | T=28 | T=32 |

Baseball pitch

Real

Prediction #1

Prediction #2

Pull ups

Real

Prediction #1

Prediction #2

Figure 5: Variety of prediction results induced by the random sampling of **z** value in the motion generator.

Figure 6: Video prediction results on UvA-NEMO and MGIF datasets.

Figure 7: Comparison of video prediction results on UvA-NEMO and MGIF datasets.