[Reviews · NeurIPS 2019]

Reviewer 1



Originality: The proposed method is an extension of the work from [20] for generating action-conditioned videos. The authors propose a modification of the landmark discovery pipeline to also generate foreground masks for the object moving in the video. Once the landmark discovery network is trained, the generated landmarks are used for supervision to train a stochastic sequence generation model. To the best of my knowledge, this is the first model to achieve this without the use of landmark labels. Quality: The paper writing needs work. There are many grammar errors throughout the paper that I noticed (e.g., line 199: detected keypoint are -> detected keypoints are). I recommend the authors to fix these in the next revision. The illustration of results are good and showcases the quality of the generated videos, and also, the supplementary material is very helpful to determine the spatio temporal quality of the generated videos. I would have, however, liked to see side-to-side video comparisons with the baselines. Clarity: The method explanation is clear, however, I feel the way the mechanical turk evaluation setup is explained could be improved. Significance: The quantitative and qualitative results given the unsupervised nature of the method makes the video generation results highly significant. In addition, the applications for video inpainting, object removal and pose translation can be of interest to researchers in the area of image/video editing

Reviewer 2



Pros: • The paper is well written and easy to follow. • The proposed approach uses only a single reference image to produce long range prediction in the future. • The model is object class agnostic, in the sense that it does not uses the class information of the foreground object to construct the key-point sequences. • The method is robust to moving background, as claimed by the authors and supported by few qualitative results. • The visual results are better than the compared approaches. • Key-point learning without any external labels appears to be a novel addition. Cons/Questions: • Even though the authors claim that the method is robust to moving background, what happens if the background contains motion from similar objects but in different directions? For example, consider a street scene where the background may contain several cars and/or pedestrians moving in all directions. • Is this method applicable for scenes with multiple foreground objects as well? These objects can belong to same or different classes. • In lines 110-115, the method hallucinates the foreground in the future based on the single reference image given. Also, the background information is carried forward from this reference input. How does the model handles situations where the foreground as well as the background is moving (for example, a sports scene such as skating or a moving camera)? In these cases, the background will not be similar to the one in the reference input in case of long-term future. • The use of a small pool of AMT human workers to rank the models is not a reproducible method for future works. Even though this is an unbiased ranking, a quantitative study involving traditional metrics like PSNR/SSIM etc. would help other models to gauge their performance against the proposed one. What is the rank of ground truth for such human evaluation ? That could have acted as a solid reference/benchmark to compare in relative scale. • Is the model end-to-end trainable? • Failure cases not discussed Also, strangely the sample videos in supple doc, have identical aliasing artifacts/degradation uniformly through all the frames. This is not consistent with more degradation as time progresses - which is evident ofcourse in frames of prediction given in paper & supple-doc. Another noticeable fact is that your prediction is not in sync with the ground truth, as far as the pose of the moving object is concerned. Is this the result of disentangling motion and content ? There has been lot of work on video frame prediction recently and many of them use the idea of disentangling motion and content. Also, it will be better to compare the model performance using auxiliary metrics such as action recognition accuracy on the produced output. Also, what about cross-dataset evaluation? In Sec. 3.2, its not clear what you trained your system for Nemo and Mgif data. Also the use of trivial adversarial loss functions used; This does not add value to CNN or GAN literature. Recent papers on video prediction have used other forms of objective functions. Why not try a few variants.

Reviewer 3



SUMMARY This paper proposes an action-conditional video prediction approach based on unsupervised keypoint detection. Given an input image and an action category label, the approach takes three steps to generate an output video: It (1) detects keypoints from an input image; (2) generates a sequence of future keypoints of the given action category; and (3) translates each keypoint frame into the target RGB image. Each step uses a dedicated neural network. The authors design a two-stage training strategy: First, the keypoint detector and the translator are trained on pairs of RGB images (in an unsupervised manner). The motion generator is then trained on video samples while fixing the other two networks. The authors demonstrate their approach on three datasets, Penn Action, UvA-NEMO and MGIF, comparing with three recently proposed approaches [12, 17, 19]. ORIGINALITY The proposed approach eliminates the need for keypoint-labeled datasets through unsupervised learning. This makes it generalizable to a variety of different scenarios where the ground-truth keypoints are not easy to obtain. This is conceptually attractive and will likely to encourage more research in this direction. The authors make two modifications to [20] for their keypoint detection network: (1) Instead of using keypoints of the target frame only, use keypoints of both input and target images; (2) Instead of predicting the target RGB image directly, predict a foreground image and a mask image, and then blended them with the input image to produce the target image. While the modifications are reasonable, it is unclear whether they actually improves the quality of keypoint detection results compared to [20]. Also, mask-based image generation is not entirely new; several existing approaches already adapt the same idea, e.g., [2] and vid2vid [Wang et al., 2018]. Overall, the proposed approach is a combination of well-known techniques for a novel scenario. The technical novelty seems somewhat incremental. QUALITY The design of the proposed approach is reasonable. However, it is unclear why the modifications to [20] were necessary; the authors do not show ablation results to justify their modifications. The approach is demonstrated only qualitatively using a limited number of examples and through a human evaluation. While the provided examples in the main paper and the supplementary look promising, it would've been far more convincing if the authors have provided more examples of different scenarios, e.g., providing the same input image with different class labels, varying the magnitude of the random vector z to show the network generating different videos, etc. At its current form, the results are not that much convincing. The UvA-NEMO dataset contains videos of smiling people with subtle differences, e.g., spontaneous vs. posed. Therefore, it is difficult to see if the results actually model the correct action class. Why not use other datasets that provide facial expressions with more dramatic differences across classes? For example, the MUG dataset contains videos from 6 categories of emotion (anger, disgust, fear, happy, sad, surprise). CLARITY The paper reads well overall, although there are a few typos and grammatical errors. I think Figures 9 and 10 are not so much informative. It would've been better if they were moved to the supplementary (with accompanying demo videos) and instead the authors showed results from an ablation study. SIGNIFICANCE Keypoint-based video prediction has become popular in the literature. This work contributes to the literature by showing the potential of unsupervised keypoint learning, eliminating the need for keypoint-labeled datasets. I like the direction of this paper and the proposed approach sounds reasonable. However, I have some concerns on insufficient experimental results.

[Author Response · NeurIPS 2019]

**(R1, R2, R3)** We will correct all typos, grammatical errors, and misleading notations (e.g. Eq. 4) in the revision. We
will also clarify ambiguous terms and cross-dataset evaluations.

**(R1) Side-to-side video comparison.** We will add side-to-side video comparison in the revision.
**(R1) User study Details.** We sampled a trajectory once for each video by randomly sampling the latent variable.
During user study, users were shown four videos side by side, and asked to rank them according to the criteria we
described in the paper. The order of four videos was randomly chosen for each vote.
**(R1, R2) More quantitative evaluation.** The table below shows more quantitative results. As our method is
stochastic, we measure the quality of generated videos using Fréchet Video Distance (FVD) instead of metrics such as
PSNR or SSIM. For the Penn action dataset, we additionally show the accuracy of human action recognition to evaluate
the performance of motion generation using the two-stream CNN. We will add this table with discussion in the revision.

| Method | Dataset | Real test data | Ours | [17] | [12] | [19] | Method | Dataset | Ours | [12] | [19] |
|---|---|---|---|---|---|---|---|---|---|---|---|
| Action recognition (%) | Penn Action | 83.33 | 68.89 | 47.14 | 40.00 | 15.55 | FVD | UvA-Nemo | 162.4 | 265.2 | 666.9 |
| FVD (lower is better) | Penn Action | - | 1509.0 | 2187.5 | 3324.9 | 4083.3 | FVD | MGIF | 409.1 | 1079.6 | 683.1 |

**(R1) Effect of adversarial loss in the motion generator.** To show the effectiveness, we measured the ac-
tion recognition accuracy of our network, and our network without adversarial loss. The results are 68.89 and
66.67 respectively, which indicates that the adversarial training is effective in generating more realistic motion.
**(R1) Visualization of learned keypoints.** Figure below shows examples of learned keypoints on each dataset where
each colored dot indicates a specific keypoint. In UvA-Nemo dataset, keypoints are usually distributed around features
of face since most of the videos have movements in local regions around the mouth and eyes. As can be seen in the
figure (leftmost), our keypoint detector is not aware of body orientation. In the example, the upper-left green point
represents "right hand" on the first sample, while it represents "left hand" on the second sample. We will add this
discussion in the revision.

Penn Action          UvA-Nemo          MGIF

**(R2) Multiple objects & Learning foreground/background.** This is a good point. Handling multiple moving
objects is still challenging for our method and all other prior methods. We showed failure examples in this case in Fig.
6 of the supplementary material. Our method also fails in generating scenes with drastic change of background since it
focuses on learning motion of a foreground object. We agree that considering multiple objects with dynamically
changing background is desirable, and we believe that our work can inspire follow-up research for learning complex
scenes. We will discuss it with more failure cases in the revision.
**(R2) End-to-end training.** Our training consists of two stages, and end-to-end training is unstable. In the future, we
plan to improve our method to be end-to-end trainable.
**(R2) Identical artifacts.** The uniform artifacts are mainly caused by imperfect mask prediction. In this case,
foreground contents in the input image are identically copied to all frames.
**(R2) Sync with the ground-truth.** As our method is stochastic, sampled videos may not sync with the ground-truth.
**(R2) Adversarial loss.** We tried advanced losses, but we found a vanilla GAN was the best in terms of visual quality.
**(R2) Evaluation of the detected keypoints.** Please note that direct evaluation is difficult as our learned keypoints do
not have ground-truth. We show the improvement of keypoint detection over the original work [20] in **(R3) Ablation
study on the image translator.** The SIFT-based dynamic features may be applicable, but we believe that our network
predicts more informative keypoints by learning proper locations to synthesize body parts from data.
**(R2) Effect of class condition.** We compare FVD scores of our method and our method without class condition, which
are $1493.2 \pm 22.9$, and $1520.2 \pm 32.7$, respectively. Even though removing class condition slightly affects the
performance, the gap is negligible compared to the baseline results.

**(R3) Ablation study on the image translator.** In Fig. 1 of the supplementary material, we compared results of (i) the
original work [20], (ii) our network (+reference keypoints, -mask), and (iii) our network (+reference keypoints, +mask)
on the Penn action dataset. Comparing (i) and (ii), our network works better on learning keypoints. We conjecture that
explicitly learning the analogical relationship of the keypoints and the images enables our network to learn the
keypoints detection easier without disentangling moving background. Comparing (ii) and (iii), our full network
produces more plausible images as the masking enforces our network to focus on synthesizing foreground object,
which is beneficial for the network to fool the image discriminator. We will add detailed discussion in the revision.
**(R3) More generation results.** We kindly remind you of the generation results of different scenarios shown in Fig. 4
and 5 of the supplementary material. We will add more results with discussion in the revision.
**(R3) The UvA-NEMO dataset.** Our purpose of using this dataset is to evaluate performance on learning class-agnostic
motion. Nevertheless, we agree with your point and we will add results on the MUG dataset in the revision.

[Meta-Review · NeurIPS 2019]

Reviewers appreciated the empirical results in this paper most of all, and the fact that they come from relatively small but well-motivated technical improvements to a well-known recent approach for unsupervised keypoint discovery. These two key changes, combined with the idea of using the discovered keypoints for long term video prediction is the main contribution in this work. Initial experiment design and evaluation in the submission was somewhat sloppy and relied on subjective scores of a small sample of video generations, but the author response to reviews now shows strong quantitative results across datasets, which are compelling. I learn marginally towards accept on this submission.